# Does Simulated Porcelain Firing Influence Corrosion Properties of Casted and Sintered CoCr Alloys?

**DOI:** 10.3390/ma14154147

**Published:** 2021-07-26

**Authors:** Dorota Rylska, Grzegorz Sokolowski, Monika Lukomska-Szymanska

**Affiliations:** 1Institute of Materials Science and Engineering, Lodz University of Technology, 1/15 Stefanowskiego St., 90-924 Lodz, Poland; dorota.rylska@p.lodz.pl; 2Department of Prosthetics, Medical University of Lodz, 251 Pomorska St., 92-213 Lodz, Poland; grzegorz.sokolowski@umed.lodz.pl; 3Department of General Dentistry, Medical University of Lodz, 251 Pomorska St., 92-213 Lodz, Poland

**Keywords:** porcelain firing, corrosion resistance, Co-Cr alloys, sintered alloys

## Abstract

The aim of the study was to evaluate how heat processing used for dental porcelain firing influences the surface properties of sintered and casted CoCr alloy. Two CoCr alloys, Soft Metal LHK (milling in soft material and sintering) and MoguCera C (casting), were used for the study. The samples were examined using SEM–EDS before and after heat treatment. Next, corrosion examinations (E_corr_, j_corr_, polarization curve, E_br_) were performed. Finally, the samples were evaluated under SEM. Based on the results, the following conclusions might be drawn: 1. Thermal treatment (porcelain firing) did not cause chemical impurities formation on the surface of CoCr alloy; 2. The sintered metal exhibited significantly higher corrosion resistance than the casted one due to its homogeneity of structure and chemical composition; 3. Heat treatment (porcelain firing) decreased the resistance of casted and sintered CoCr alloy to electrochemical corrosion. The reduction in corrosion resistance was significantly higher for the casted alloy than for the sintered alloy; 4. The corrosion resistance decrease might be due to an increased thickness and heterogeneity of oxide layers on the surface (especially for the casted alloy). The development of corrosion process started in the low-density areas of the oxide layers; 5. The sintered metal seems to be a favourable framework material for porcelain fused to metal crowns.

## 1. Introduction

Despite an increasing popularity of all-ceramic reconstructions, porcelain fused-to-metal (PFM) crowns remain the gold standard in fixed prosthodontics. This is due to the combination of aesthetics (provided by porcelain) and strength (ensured by metal alloy). The metal framework can be made from noble (i.e., gold alloys) or base metal alloys (i.e., cobalt–chromium, nickel–chromium alloys). Metallic materials in the oral cavity are subjected to galvanic pitting, crevice, fritting, fatigue corrosion and, eventually, stress corrosion cracking (SCC) [1]. The corrosion of metals and alloys may be caused by chloride ions, high oxidation level and a relatively high temperature (370 °C) of the electrolyte solution present in the oral cavity. The ever-changing conditions in the mouth (e.g., changes in saliva pH, the presence of food and microorganisms) exert a highly unfavourable impact on these materials. Unfortunately, cobalt–chromium (CoCr) and nickel-chromium (NiCr) alloys are not exempt from corrosion processes [2,3,4,5].

Corrosion may attack the surface of a metal framework of PFM crowns that is exposed to saliva. This process might cause ion release [6]. Moreover, similar phenomena occur in the gap at the metal–porcelain interface. However, corrosion mostly affects the inner surface of PFM crowns, as none of the cements can provide a perfect seal. This allows electrolytes to penetrate and stay in contact with the inner metal surface, which causes crevice corrosion. This process influences biocompatibility and clinical performance of the restoration [1].

The alloy’s structure and surface properties mostly determine its corrosion resistance. A coarse-crystalline structure, dendritic micro-segregations and exuded coarse-grained carbides are the main structural defects of the casted CoCr alloys [7,8]. Although casting is time consuming and requires qualifications, it remains the dominant method in metal processing. Even so, several alternatives to the traditional casting of CoCr frameworks have been introduced [9]. They include milling in solid metal, milling in soft material and sintering (MSM) or selective laser melting (SLM). Subtractive manufacturing processes (i.e., milling) provide a reduction in pores and defects, but their potential for a complex construction fabrication is limited, as compared with casting and SLM. Milling in solid metal relies on manufacturing of homogeneous alloy blocks under standardized industrial conditions [10,11]. MSM, in contrast to SLM, does not require any expensive equipment. This technique is accompanied by subsequent sintering in a protective atmosphere, i.e., argon, that enables manufacturing of frameworks exhibiting homogenous structure and suitable for almost each porcelain material [12,13,14]. SLM is a rapid prototyping technique, 3D printing or additive manufacturing (technology that grows three-dimensional objects by one superfine layer at a time). It uses high density laser for melting and fusion of metal powders in and between layers. SLM technique allows for manufacturing elements of different relative density (up to 99.9%). It also minimizes the probability of error during production and the number of possible defects, provides high homogeneity of the structure and is waste free. Studies show that SLM frameworks are of better quality than the casted ones [15,16,17,18,19,20,21].

The research on CoCr dental alloys layered with porcelain focus on the evaluation of mechanical properties and porcelain–metal bond strength [22,23], the influence of surface treatment on the surface roughness [24] and the influence of different factors on the initial stage of oxidation (before porcelain layering) [25,26,27]. The influence of porcelain firing on the corrosion of dental alloys has not been completely clarified yet. Procedures of porcelain firing to metal cause metal surface oxidation and the formation of mill scale, which may cause corrosion, and therefore impair the biological properties of the material [28,29,30,31,32,33]. The literature on milling in soft material with subsequent sintering concentrates mostly on fusion to porcelain [22,34,35,36,37] with scarce research on corrosion resistance [38,39,40].

Therefore, the aim of the study was to evaluate how heat processing used for dental porcelain firing influences the surface properties of sintered and casted CoCr alloy.

## 2. Materials and Methods

Two CoCr alloys, Soft Metal LHK and MoguCera C, were used for the study. The chemical composition of alloys is shown in Table 1.

### 2.1. Sample Preparation

The soft metal specimens were prepared from fabricated powder blocks (CoCr alloy combined with a binding agent). The samples were fabricated by means of milling and subsequent sintering (MSM). The specimens’ shapes were designed using CAD Autodesk 123D Design software, version 2.1.11 (Autodesk Inc., San Rafael, CA 94903, USA). The material was formed into cylinders (diameter of 14 mm, height of 8 mm) by means of dry processing with dust extraction, using a four-axis numerical milling machine CNC Amann Girrbach Ceramill Motion and CAM computer software (Amman Girrbach, Pforzheim, Germania). Afterwards, the cylinders were sintered to full density in Amann Girrbach Ceramill Argotherm 2 furnace in argon atmosphere (Amman Girrbach AG, Koblach, Austria).

The samples of MoguCera C alloy were casted using the lost-wax/lost-resin technique. The moulds, pre-designed in CAD software, were printed with a Form 2 3D printer (3D printing technology—SLA, Formlabs, Sommerville, MA, USA) using photopolymer resin dedicated for producing casts from Castable Resin (Formlabs, USA). Subsequently, a mould using Bellavest SH investment material (Bego, Bremen, Germany) was fabricated. The mould was subjected to shock heating in final temperature of 930 °C. The cast was produced in the Pi Dental Silver Cast device (Pi Dental, Budapest, Hungary), utilizing electromagnetic induction for alloy melting and the centrifugal force for pressing it into the investment. The study design is presented in Table 2.

### 2.2. The Heat Treatment Simulating the Porcelain Firing

Cylindrical samples were prepared with abrasive paper (grit size 180, 320, 600, 800, 1200, 2400) and the frontal upper surface was polished with diamond slurry (3 μm) using a Minitech 233 (Presi, Le Locle, Switzerland) polishing machine. After degreasing and steam-cleaning, the samples were subjected to the metal-porcelain firing cycle under vacuum in a dental porcelain furnace (Programat EP3000, Ivoclar Vivadent Amberst, NY, USA). The protocol for parameters used for opaquer, dentin and glaze firing simulation is given in Table 3.

### 2.3. Sample Evaluation

Samples were investigated as presented in Table 2. Samples without heat treatment served as a control group.

#### 2.3.1. SEM–EDS Study

The morphology of the samples was evaluated with a Secondary Electron (SE) and Backscattered-Electron (BSE) detector of JEOL JSM-6610LV scanning microscope (JEOL, Peabody, MA, USA). The accelerating voltage of 20 keV was used. The metallographic study of 3 samples in each study group was performed. The investigation was carried out in several magnifications ranging from ×500 to ×2500.

The EDS method was used to qualitatively and quantitatively analyze the chemical composition and changes of the micro-areas of the samples. Moreover, the effect of heat treatment on each alloy was assessed. The microanalysis of the chemical constitution was performed with an EDS X-MAX 80 microanalyzer (Oxford Instruments, Abingdon, UK). Three samples from each study and the control group were evaluated.

#### 2.3.2. Corrosion Examinations

The specimens were degreased, rinsed with ethanol and dried. The electrochemical analysis was performed in a Radiometer-Analytical electrochemical cell (CEC/TH Thermostated Multipurpose Cell—Radiometer Analytical, Lyon, France). The working electrode (Ew) being the studied specimen, the platinum counter electrode (Ec) and the calomel reference electrode (Eref) were used for the study. Each sample’s working surface was ca. 0.95 cm^2^. All the samples were exposed to the solution for 2 h to establish an open circuit potential (OCP). The Tafel extrapolation method in the domain of the ±200 mV, SCE vs. the OCP was used to determine the value of the corrosion current density jcorr and the corrosion potential Ecorr during studying the first sequence. Then, the cathodic and anodic polarization, in full voltage spectrum (from −1.0 V to 1.5 V), using the potentiodynamic method at the speed of potential changes of 1 mV/s, was executed. The potentials for abrupt current increase (E_br_) were also estimated from the polarization curve, as the potential of the passive film breakdown caused by pitting or by transpassive dissolution. The E_br_ value was estimated for the inflection point from the polarization curves. There have been different approaches in literature concerning the evaluation of the breakdown potential E_br_, according to which it could be determined at the inflection point [41,42], associated with a sharp increase in current or as the potential at which the rising current permanently exceeded 10 μA/cm^2^ [43]. The polarization curve was determined for each sample once.

The measurements were performed in a 0.9% NaCl solution at room temperature. ATLAS 0531 Electrochemical Unit & Impedance Analyser (Atlas–Sollich, Rębiechowo, Poland) with the AtlasCorr05 software, version 3.19 (Atlas–Sollich, Gdansk, Poland) was applied. The calculations to determine the corrosion process parameters were performed using AtlasLab software version 2.9(Atlas–Sollich, Gdansk, Poland).

## 3. Results

### 3.1. SEM–EDS Study

#### 3.1.1. Samples without Heat Treatment and before Corrosion Examination

The comparison of microstructure (SEM) and composition (EDS) for both metals revealed significant differences. The SM-Control samples showed grains of various sizes, but of a uniform chemical composition, with pores present mostly at grains’ boundaries and less numerous inside the grains (Figure 1a). The SM-Control samples presented a uniform distribution of Co, Cr and Mo (Figure 1b). The MC-Control samples were characterised by a typical dendritic structure of austenitic matrix with interdendritic carbides, intermetallic phases of irregular size and pores (Figure 2a). The MC-Control exhibited a non-uniform distribution of Co, Cr and Mo (Figure 2b). The first two elements accumulated mostly in the austenitic matrix, while Mo gathered in the areas of interdendritic precipitates (Figure 2b).

#### 3.1.2. Samples after Heat Treatment and before the Corrosion Examination

The EDS analysis (Figure 3 and Figure 4) of both alloy samples allowed for assessing the surface state. There were no impurities in the MC-HT and SM-HT in the EDS study. Maps of surface distribution of elements for the SM-HT (Figure 3b) showed a more uniform distribution of oxygen and Cr compared to the MC-HT (Figure 4b). Figure 5 and Figure 6 reflect the morphology of the oxide layers of both sample types with varied local cracks. These results are consistent with the results of EDS analyses.

#### 3.1.3. Samples before Heat Treatment and after the Corrosion Examination

The signs of corrosion on the grain boundaries in the areas of pores were present on the surface of the SM-Control-Corr (Figure 7). It was visible that the depth of the holes increased after corrosion tests. However, no sign of galvanic corrosion was observed on the surface with uniform structure.

Studies of the MC-Control-Corr (Figure 8) also demonstrated the presence of pores and corrosion pits; the number of pits and pores was significantly higher than for the SM-Control-Corr. The surface damage (corrosion pits) on the samples in both study groups probably resulted from the wide potential range applied.

#### 3.1.4. Samples after Heat Treatment and after the Corrosion Examination

The signs of corrosion were only visible in the grain boundary areas and around the pores for the SM-HT-Corr (Figure 9). Some signs of preferential dissolution at the interface between the matrix dendrites and the precipitates could be observed in the MC-HT-Corr (Figure 10). Moreover, pits and deep cracks of thick oxide layers were visible on the surface of the MC-HT-Corr specimen (Figure 10). The MC-HT-Corr oxide layers with a diverse composition and thickness suffered a more intense cracking than the SM-HT-Corr.

### 3.2. Corrosion Examination

Table 4 and Table 5 present the values of the corrosion current density (j_corr_) and the corrosion potential (E_corr_) determined by means of the Tafel straights extrapolation, for the control and the HT samples, respectively.

MC-Control and MC-HT presented higher values of jcorr and lower Ecorr (Table 3 and Table 4, respectively) in comparison to both Soft Metal samples.

Potentiodynamic characteristics for polarization between −1.0 V and 1.5 V at potential change rate of 1.0 mV/s are shown in Figure 11.

The potentiodynamic curves for the SM-Control and the SM-HT (control and HT) (Figure 11) were shifted in the positive direction, anode current densities were lower than for the MC-Control and the MC-HT. For the MC-HT, the shift of the potentiodynamic curve to the left and a marked increase in anodic current values were observed.

The SM-Control, MC-Control and SM-HT showed a passivity in a relatively wide range of potentials in which the current density slightly increased linearly until a breakdown potential was reached. The part of the polarization curve above the breakdown potential (E_br_) for the SM-Control (E_br_ = 0.785 V) and the MC-Control (E_br_ = 0.710 V) was characterized by an increase in the current due to the dissolution of the protective oxide films and water oxidation (as the potential of the oxygen-evolution in the neutral saline solution was about ~0.6 V). The value of E_br_ for the SM-HT was equal to 0.560 V. There was no evidence of abrupt increase of current for the MC-HT, but the anodic part of the polarization curve presented high values of current densities (~1–2·× 10^−3^ A/cm^2^).

## 4. Discussion

In this study, Soft Metal (MSM and sintering) exhibited a homogenous structure, on the contrary to MoguCera C (casting). Similar results were obtained by other researchers [34,38,39,44,45]. The highly inhomogeneous structure of MorguCera C resulted in an increased susceptibility to corrosion, due to a less stable passive oxide layer on the surface; as observed by other authors as well [16,46].

Similarly, after HT, there were no changes in the structure homogeneity/inhomogeneity and in the chemical composition of the superficial layers of both alloys, except for the appearance of oxygen. The present findings are supported by literature [45]. Moreover, SM-HT exhibited a more homogenous structure and chromium and oxygen distribution. Conversely, MorguCera C exhibited inhomogeneity and a relatively low corrosion resistance. This surface characteristics resulted in an increased susceptibility to corrosion when compared to Soft Metal alloy (both before and after HT). It should be emphasized that the relationship between the structure inhomogeneity and an increased susceptibility to corrosion has been well documented [7,29,46,47,48,49].

In the present study, the SEM–EDS results were confirmed by the corrosion examination. The parameters describing the corrosion properties (corrosion current density and corrosion potential), obtained by the extrapolation of the Tafel straights, revealed a higher corrosion resistance of the sintered material (both in the control and the HT group). Similarly, the pitting corrosion resistance was found to be higher for the sintered alloy (Soft Metal), based on the polarization curve examinations. The sintered material was therefore more resistant to corrosion due to its chemical composition and uniform morphology. Similar findings have been obtained by other authors [39].

However, the influence of HT on the susceptibility to corrosion clearly demonstrated that, for both fabrication methods, corrosion resistance decreased after heat-processing simulations, although its decrease was insignificant for specimens obtained by MSM technology (Soft Metal). The possible cause of the decreased corrosion resistance after HT (MC-HT) could be attributed to the inhomogeneous composition of oxide layers. It undoubtedly led to an increased failure (presence of local cracks) and a decreased solidity, which facilitated the penetration of the corrosive environment. Moreover, the stress generated in the oxide layers increased with the layer thickness, resulting in the deterioration of protective properties of these layers.

It is worth mentioning that there has been little research on the effect of porcelain firing on the corrosion resistance of dental alloys. The chemical composition of each alloy influenced its corrosion properties both before and after HT. The CoCr alloy exhibited a significantly higher corrosion resistance (corrosive behaviour and surface quality) before and after porcelain firing in comparison to NiCr alloy. Additionally, NiCr (Be free) alloy exhibited a significantly decreased corrosion resistance after firing, while it remained unchanged for the alloys containing Be [32]. Moreover, HT-related changes in the microstructure and microhardness of the tested alloys and an increase in Co and Ni ion release were noted. A slight increase in CrO(x) on the surface of the Be-free alloy and an increased MoNi was observed on the surface of both alloys, which might be one of the reasons why Ni and Mo ion release increased after firing [32]. Other studies also confirmed a significant increase in Cr, Ni and Mo ion release after porcelain firing and changes in alloy microstructure [30].

Only a few studies have assessed the corrosion resistance of CoCr alloys [4,35,50]. The corrosion currents and the polarization resistance values of CoCr alloy used for PFM crowns fabrication were comparable to these obtained for PdAg alloys. The corrosion potentials of the CoCr alloys were lower than these of the Pd-based alloys, but the corrosion currents and the polarization resistance values were similar for both groups of alloys. However, in this study the conditions of porcelain firing were not simulated and changes in the microstructure and corrosion resistance were not evaluated [4]. Moreover, it was observed that the firing temperature of 980 °C reduced the ion release from CoCr alloy. This finding indicated the influence of HT on corrosion qualities [50]. However, these results were not supported by Xin et al. who stated similar corrosion behaviour of CoCr SLM specimens before and after porcelain firing [18].

Additionally, it was found that porcelain firing had a detrimental effect on the corrosive properties of NiCr, CoCr and PdAg alloys [51]. The most important reason why the decrease in the corrosion resistance of both tested types of alloys occurred could be attributed to an increased thickness of the oxide layers formed after HT. The increase in corrosion rates corresponded well with the reduced Cr and Mo levels in the surface oxides of the fired alloys. These results were in consistence with other study, reporting that repeated firings decreased corrosion resistance of CrCo and CrNi alloys [5]. The above-mentioned studies support the results of the present paper [5,51]. Furthermore, the corrosion resistance of CoCr alloy in various media (0.1 N NaCl, 1% citric acid and artificial saliva) increased after HT at temperature ranging from 650 °C up to 850 °C. However, an increase in corrosion rate was observed after HT at temperature of 950 °C [52]. The latter was comparable to the temperature range applied in the present study.

The most important cause of the decrease in the corrosion resistance of both alloys tested was an increased thickness of the oxide layers after HT. Different oxidation kinetic of the alloy elements caused differences in the oxide composition. It could explain the significantly lower corrosion resistance of MC-HT presenting the heterogeneity of the “primary” alloys’ structure and the oxide layers.

Moreover, during the polarization tests in a wide potential range (up to 1.5 V), the potential increased over the oxygen evolution potential. In the presence of oxygen evolution, the local pH decreased and the thickness of the oxide layers increased. These results were supported by another study [53]. A Ti-6A1-4V alloy after thermal oxidation (500, 600, 700 and 800 °C) at heating rate of 5 °C/min was evaluated. The decrease in corrosion resistance was observed after oxidation in 800 °C. This was caused by the formation of cracks and pores in thick oxide layer and the appearance of galvanic corrosion. A corresponding cause of a decrease in corrosion resistance was observed in the present study, especially for the casted alloy (MoguCera C).

The increased corrosion resistance of sintered metal before and after HT indicated that milled and sintered frameworks exhibited enhanced biocompatibility. This was due to the fact that alloy corrosion resistance depended directly on the inner structure, on the passive layers (presence and composition) and on the fabrication method [54,55].

In order to avoid a decrease in mechanical and biological properties, there has been an ongoing search for new metal alloys with a more favourable chemical composition and an increased corrosion resistance. Moreover, novel fabrication methods of metal prosthodontic frameworks have been introduced, aiming at increasing corrosion resistance and therefore enhancing biological properties with maintained or even improved mechanical properties of the alloys.

Some limitations of the present study should however be noted. The study evaluated only two alloys, processed using two dedicated methods. A wider range of materials and methods should be thus investigated. Furthermore, the composition of the sintered alloy did not fully match the composition of the casted alloy. Furthermore, the application of the electrochemical impedance spectroscopy (EIS) to examine corrosion properties should be recommended in the future research. The EIS method would provide much more accurate information on the corrosive behaviour, and would also allow for a more detailed examination, namely the analysis of the influence of porosity on the surface behaviour in the oral cavity environment. Additionally, tests after long immersion in a corrosive medium should allow for the assessment of corrosion products under OCP conditions. In the present study, potentiodynamic tests (accelerated tests) were performed. Again, further research is needed to carry out a detailed examination of the oxide layer structures and their composition (i.e., X-ray photoelectron spectroscopy).

## 5. Conclusions

Within the limitations of the study, the following conclusions might be drawn:Thermal treatment (porcelain firing) did not cause chemical impurities formation on the surface of CoCr alloy.The sintered metal exhibited significantly higher corrosion resistance than the casted one due to its homogeneity of structure and chemical composition.Heat treatment (porcelain firing) decreased the resistance of the casted and sintered CoCr alloy to electrochemical corrosion. The reduction in corrosion resistance was significantly higher for the casted alloy than for the sintered alloy.The decrease in corrosion resistance might be due to an increased thickness and the heterogeneity of the oxide layers on the surface (especially for the casted alloy). The development of corrosion process started in the low-density areas of the oxide layers.The sintered metal seems to be a favourable framework material for porcelain fused to metal crowns.

## Figures and Tables

**Figure 1 materials-14-04147-f001:**
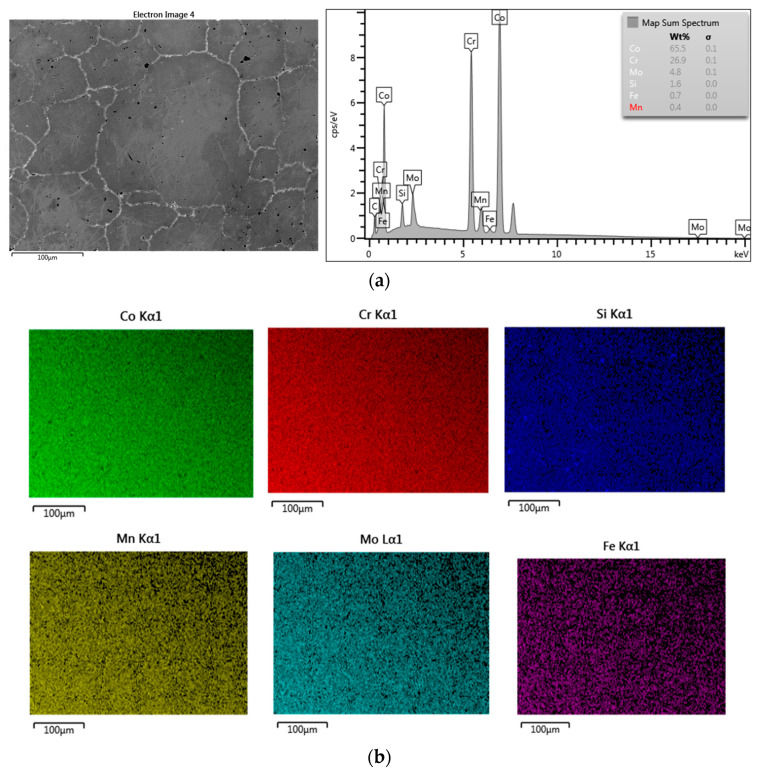
SEM–EDS results for SM-Control: (**a**) SEM-SE micrograph showing morphology with averaging EDS results from the area presented; (**b**) the surface distribution of Co, Cr, Si, Mn, Mo, Fe.

**Figure 2 materials-14-04147-f002:**
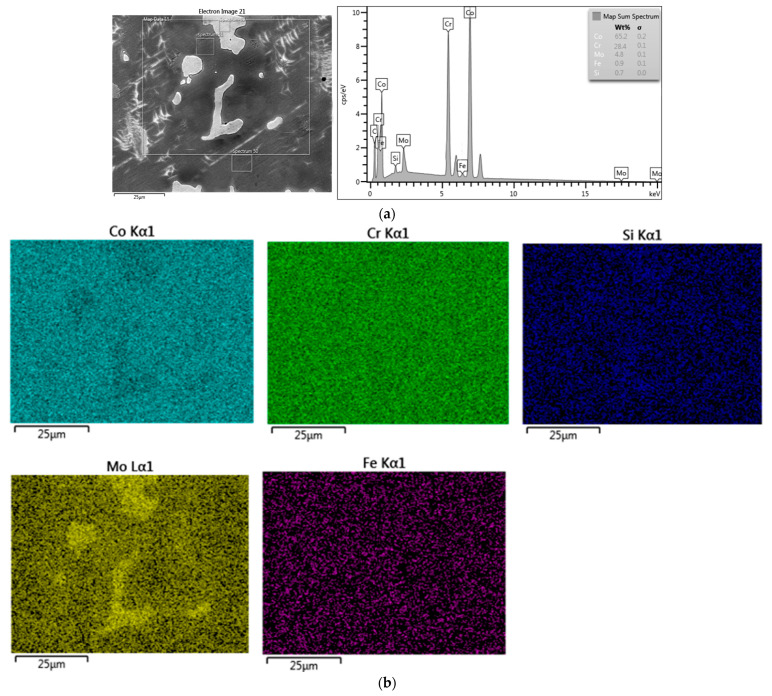
SEM–EDS results for MC-Control: (**a**) SEM-SE micrograph showing morphology with averaging EDS results from the marked area; (**b**) the surface distribution of Co, Cr, Si, Mo, Fe from the marked area.

**Figure 3 materials-14-04147-f003:**
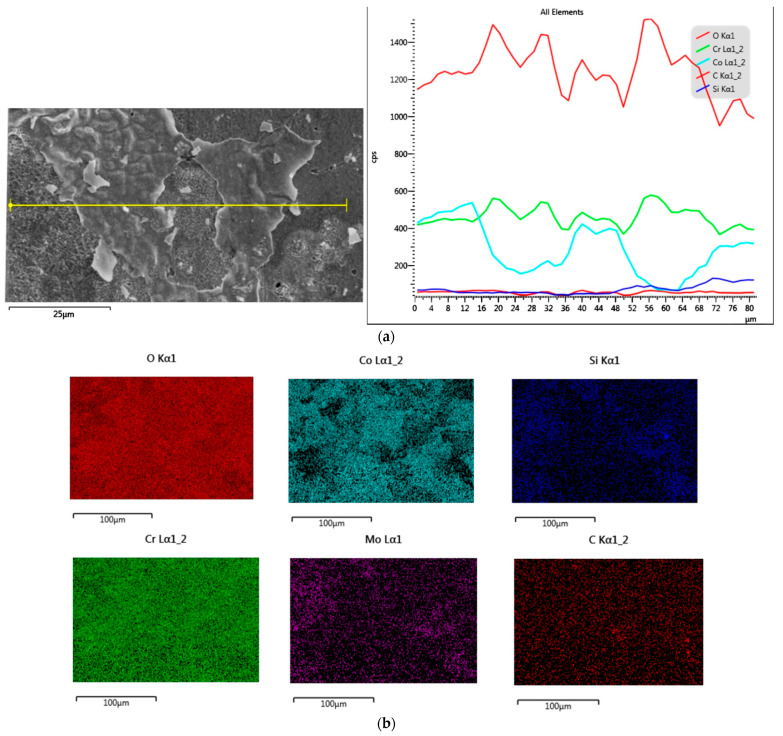
SEM–EDS results for sintered SM-HT: (**a**) SEM-SE micrograph with corresponding EDS line on the right; (**b**) the surface distribution of O, Co, Si, Cr, Mo, C; (**c**) EDS results from marked micro-areas.

**Figure 4 materials-14-04147-f004:**
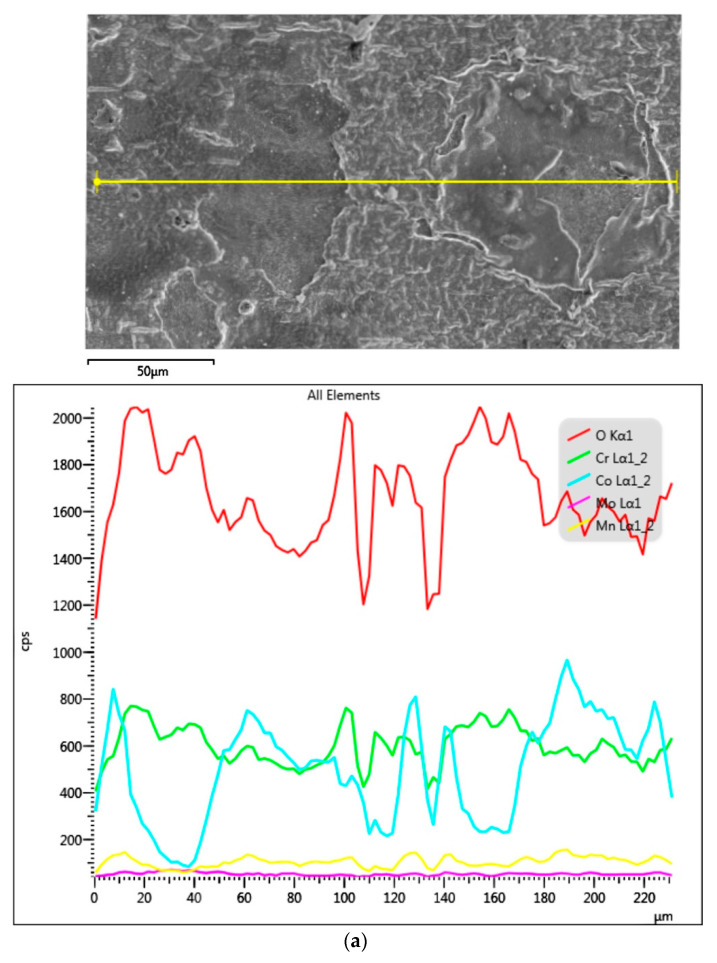
SEM–EDS results for casted MC-HT: (**a**) SEM-SE micrograph with corresponding EDS line on the right; (**b**) the surface distribution of O, Co, Cr, Mo, C; (**c**) EDS results from marked micro-areas.

**Figure 5 materials-14-04147-f005:**
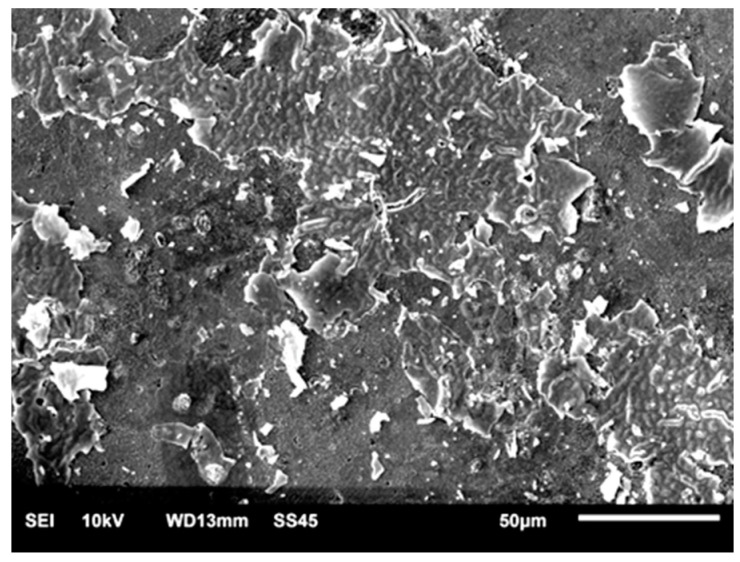
SEM-SE surface morphology of SM-HT (×500).

**Figure 6 materials-14-04147-f006:**
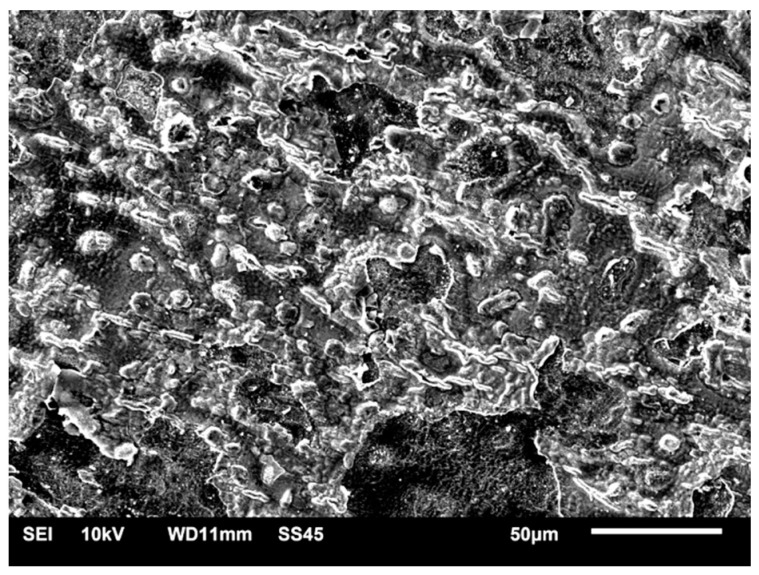
SEM-SE surface morphology of casted MC-HT (×500).

**Figure 7 materials-14-04147-f007:**
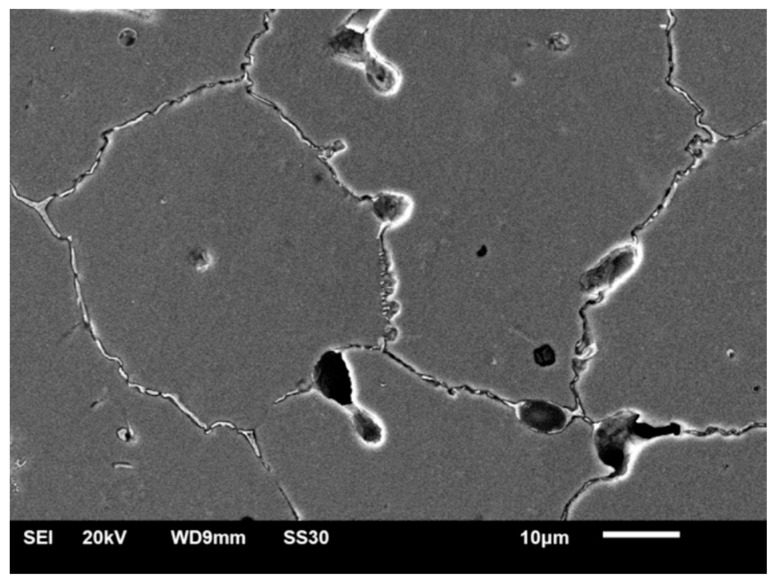
SEM-SE surface morphology of SM-Control-Corr (×1300).

**Figure 8 materials-14-04147-f008:**
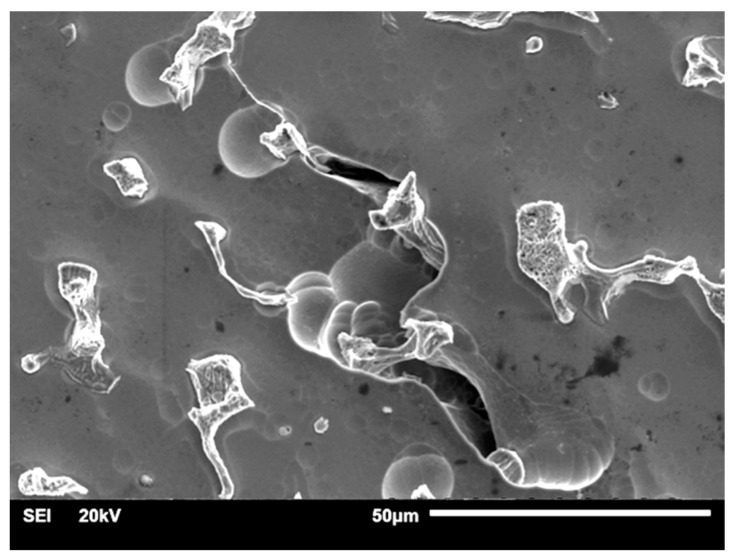
SEM-SE surface morphology of MC-Control-Corr (×1000).

**Figure 9 materials-14-04147-f009:**
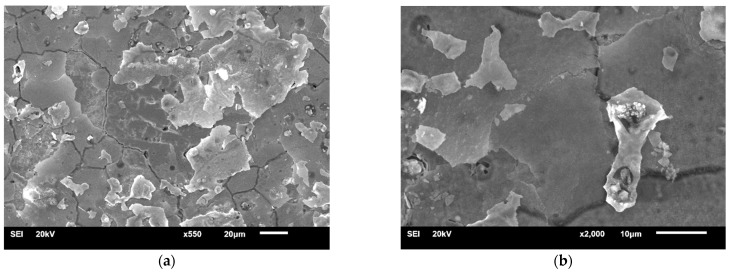
Morphology of SM-HT-Corr: (**a**) ×550; (**b**) ×2000.

**Figure 10 materials-14-04147-f010:**
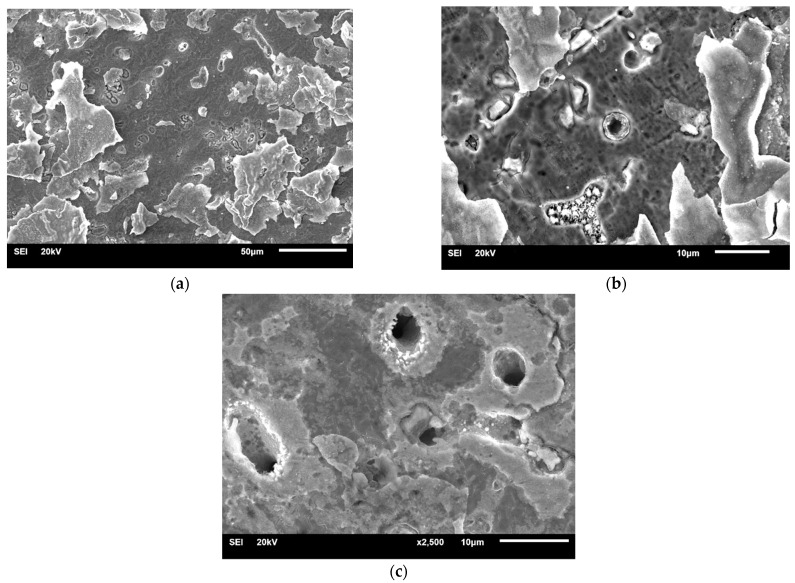
Morphology of MC-HT-Corr: (**a**) ×550; (**b**) ×2000, (**c**) ×2500.

**Figure 11 materials-14-04147-f011:**
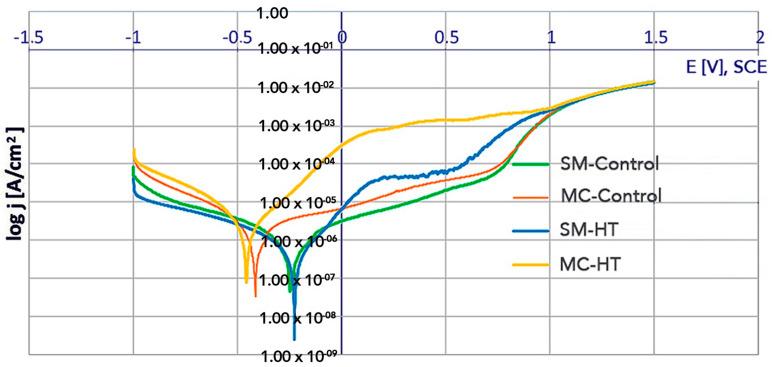
Potentiodynamic polarization curves in 0.9% NaCl solution.

**Table 1 materials-14-04147-t001:** The chemical composition of alloys (wt.%).

Material	Company (Country)	Method	Co	Cr	Mo	Si	Mn	Other Elements
Soft Metal	LHK (Jicheon-myeon, Korea)	MSM	63.4	29	5.8	max 1	-	<1%
MoguCera C	Scheftner (Mainz, Germany)	casting	65	28	5	Si + C < 1%	1	-

MSM: milling in soft material and sintering.

**Table 2 materials-14-04147-t002:** Study design.

Sample Material	Soft-Metal (SM)		MoguCera C (CM)	
Manufacturing method	milling in soft material + sintering		casting	
Heat treatment	-	+	-	+
Study groups	SM-Control (n = 3)	SM-HT (n = 3)	MC-Control (n = 3)	MC-HT (n = 3)
Evaluation	SEM–EDS			
	Corrosion examinations (E_corr_, j_corr_, polarization curve, E_br_)			
Study groups	SM-Control-Corr (n = 3)	SM-HT-Corr (n = 3)	MC-Control-Corr (n = 3)	MC-HT-Corr (n = 3)
Evaluation	SEM			

Control: sample without heat treatment; HT: sample after heat treatment; Corr: sample after corrosion examination; SEM: Scanning Electron Microscope; EDS: Energy Dispersive Spectroscopy.

**Table 3 materials-14-04147-t003:** Firing simulation parameters for porcelain.

Type of Porcelain Material	Opaquer	Dentin	Glaze
Initial temperature B (°C)	400	600	650
Time for drying and furnace chamber closing S (min)	8	7	5
Speed of temperature’s increase t_1_ (°C/min)	65	45	50
Firing temperature T (°C)	1000	940	910
Warming up time H (min)	1	-	-
Temperature of vacuum activation V_1_ (°C)	400	600	-
Temperature of vacuum deactivation V_2_ (°C)	1000	930	-

**Table 4 materials-14-04147-t004:** Values of j_corr_ and E_corr_ from electrochemical examination.

Sample	Sample Number	j_corr_ 10^−6^ [A/cm^2^]	E_corr_ [mV]
MC-Control	I	5.012	−367.11
	II	4.954	−359.86
	III	5.032	−374.23
Average value		4.999 ± 0.041	−367.07 ± 7.18
SM-Control	I	0.943	−185.31
	II	0.932	−188.12
	III	0.975	−179.46
Average value		0.950 ± 0.022	−184.30 ± 4.42

**Table 5 materials-14-04147-t005:** Values of j_corr_ and E_corr_ from electrochemical examination.

Sample	Sample Number	j_corr_ 10^−6^ [A/cm^2^]	E_corr_ [mV]
MC-HT	I	8.977	−389.39
	II	9.021	−387.41
	III	9.074	−385.88
Average value		9.024 ± 0.049	−387.56 ± 1.76
SM-HT	I	1.023	−199.1
	II	1.034	−201.17
	III	1.046	−199.47
Average value		1.034 ± 0.011	−199.91 ± 1.10

## Data Availability

Data Sharing is not applicable.

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
