# Peer review of "Does Simulated Porcelain Firing Influence Corrosion Properties of Casted and Sintered CoCr Alloys?"

_materials, 2021, doi:10.3390/ma14154147_

Round 1

Reviewer 1 Report

Manuscript ID - materials-1294898

Reviewer’s comments - Does simulated porcelain firing influence corrosion properties of casted and sintered CoCr alloys?

The introduction has been well written, comprehensively explains the background to the study and shows the gap in the current knowledge regarding the topic. The aims of the study have been clearly stated.

The method section has been well described and the methodology adopted will produce valid and reliable results.

The results are clearly presented and well explained.

The quality of the discussion was good, the results of the study was compared and related to previous studies. The limitations of the study were appropriately addressed.

The conclusions were valid and based on the results of the study.

Overall, this is a well done study with a novel finding.

General comments.

The definite article words such as “a” and “the” are missing in many of the sentences, eg., line 30 should read – “Metallic materials in the oral cavity are…”  Line 35 should read – “exert a highly unfavourable impact on these….” Line 38 should read – “Corrosion may attack the surface of a metal framework of PFM crowns…..” The manuscript needs to be checked for English grammar errors.

In several places the manuscript has been written in the present tense. It should be written in the past tense.

Line 29. The use of the terms “precious” and “non-precious” is technically and academically incorrect. The terms refer to the monetary value of the alloys and not their constituents or type. The correct terminology to describe dental alloys is “noble” or “base”. This needs to be corrected. See the definition of alloy in the Glossary of Prosthodontic Terms - https://www.academyofprosthodontics.org/lib_ap_articles_download/GPT9.pdf

Line 105. To which surface was the polishing protocol applied? As it reads, it implies that the polishing was applied to all surfaces. This needs to be clarified.

Author Response

Dear Sir or Madam,

Thank you for your review. We greatly appreciate the important suggestions we have received. The text has been extensively revised based on your recommendations. Point-by-point responses are below and modifications in the manuscript were highlighted in track-change in Word.

Reviewer’s comments - Does simulated porcelain firing influence corrosion properties of casted and sintered CoCr alloys?

 The introduction has been well written, comprehensively explains the background to the study and shows the gap in the current knowledge regarding the topic. The aims of the study have been clearly stated.

 The method section has been well described and the methodology adopted will produce valid and reliable results.

 The results are clearly presented and well explained.

 The quality of the discussion was good, the results of the study was compared and related to previous studies. The limitations of the study were appropriately addressed.

 The conclusions were valid and based on the results of the study.

 Overall, this is a well done study with a novel finding.

R: Thank you for your kind words.

 General comments.

 The definite article words such as “a” and “the” are missing in many of the sentences, eg., line 30 should read – “Metallic materials in the oral cavity are…”  Line 35 should read – “exert a highly unfavourable impact on these….” Line 38 should read – “Corrosion may attack the surface of a metal framework of PFM crowns…..” The manuscript needs to be checked for English grammar errors.

 In several places the manuscript has been written in the present tense. It should be written in the past tense.

R: Thank you for your comment. The manuscript was revised and corrected following your recommendations.

 Line 29. The use of the terms “precious” and “non-precious” is technically and academically incorrect. The terms refer to the monetary value of the alloys and not their constituents or type. The correct terminology to describe dental alloys is “noble” or “base”. This needs to be corrected. See the definition of alloy in the Glossary of Prosthodontic Terms - https://www.academyofprosthodontics.org/lib_ap_articles_download/GPT9.pdf

R: Thank you for your comment. The manuscript was revised and corrected following your recommendations. Now the sentence reads (lines 28-30): The metal framework can be made from noble (i.e. gold alloys) or base metal alloys (i.e. cobalt-chromium, nickel-chromium alloys).

 Line 105. To which surface was the polishing protocol applied? As it reads, it implies that the polishing was applied to all surfaces. This needs to be clarified.

R: Thank you for your comment. The manuscript was revised and corrected following your recommendations. The following text was added (lines 106-107): Cylindrical samples were prepared with abrasive paper (grit size 180, 320, 600, 800, 1200, 2400) and the frontal upper surface  was polished with diamond slurry…

Reviewer 2 Report

Review on Rylska et al: Does simulated porcelain firing influence corrosion properties of casted and sintered CoCr alloys?

The study is well designed, and it provides understandable description of the problem. Though the question implicit in the title is not directly described, but the study also provides answers for other questions rising from the experimental results. It also has a direct practical conclusion: which methods to use for porcelain fused to metal crowns to increase their survival.

Author Response

Dear Sir or Madam,

Thank you for your review. We greatly appreciate the important suggestions we have received. The text has been extensively revised based on your recommendations. Point-by-point responses are below and modifications in the manuscript were highlighted in track-change in Word.

Review on Rylska et al: Does simulated porcelain firing influence corrosion properties of casted and sintered CoCr alloys?

The study is well designed, and it provides understandable description of the problem. Though the question implicit in the title is not directly described, but the study also provides answers for other questions rising from the experimental results. It also has a direct practical conclusion: which methods to use for porcelain fused to metal crowns to increase their survival.

R: Thank you for your comment and kinds words. The following conclusion was added according to your recommendation (lines 377-378): 5.              The sintered metal seems to be a favourable framework material for porcelain fused to metal crowns.